# The Dynamical Origin of the Graviton Mass in the Non-Linear Theory of Massive Gravity

**Ivan Arraut** [1,2,3]

1    The Open University of Hong Kong, 30 Good Shepherd Street, Homantin, Kowloon, Hong Kong, China;
     ivanarraut05@gmail.com
2    Department of Physics, Faculty of Science, Tokyo University of Science, 1-3, Kagurazaka, Shinjuku-ku,
     Tokyo 162-8601, Japan
3    State Key Laboratory of Theoretical Physics, Institute of Theoretical Physics, Chinese Academy of Science,
     Beijing 100190, China

**Abstract:** We compare the standard Higgs mechanism corresponding to the scalar field, with the dynamical origin of the graviton mass inside the scenario of the dRGT theory of massive gravity. We demonstrate that the effective mass perceived locally by different observers depends on how they define the local time with respect to the preferred notion of time defined by the Stückelberg function $T_0(r,t)$.

**Keywords:** graviton mass; Higgs mechanism; Stückelberg function; time-direction; massive gravity

## 1. Introduction

In particle physics, for the linear $\sigma$-model, the dynamical origin of the mass for a scalar field is explained by the Higgs mechanism [1,2]. In the standard $\sigma$-model, the vacuum can be single or degenerate depending on the relation between the two free-parameters of the theory. When we have degeneracy, then the multiplicity of vacuums are connected to the existence of gapless particles (Nambu-Goldstone bosons) [3,4]. The gapless Nambu-Goldstone bosons move along the degenerate vacuum without any effort at the infrared level. When we consider gauge symmetries, the Nambu-Goldstone bosons are then eaten-up by the gauge fields which become massive. Since we are considering the possibility of having a massive graviton field, then we expect a possible formulation of a gravitational version of the Higgs mechanism. The possibility of having a gravitational version of the Higgs mechanism has been explored in [5,6]. However, the way in which this mechanism must be formulated for the case of gravity is not yet clear at all. Massive gravity is in essence a $\sigma$-model [6–8]; then the vacuum solutions in massive gravity are also degenerate for a given combination of the two free-parameters appearing inside the massive action [9]. This means that the symmetry of the theory can be spontaneously broken for some combination of parameters. Then the theory has in principle three Nambu-Goldstone bosons which are eaten-up by the dynamical metric in order to become massive. Note that in a healthy theory, the number of degrees of freedom in massive gravity is always five, namely, three Nambu-Goldstone bosons plus the two degrees of freedom for the mass-less spin-2 graviton field. After the graviton mass is generated dynamically, we have a single massive graviton field with five degrees of freedom. The breaking of the symmetries of the theory spontaneously, cannot be interpreted classically. If the results are interpreted classically from the perspective of Einstein gravity, then the degeneracy of the vacuum solutions in massive gravity is equivalent to a loss of predictability of the theory [9]. In massive gravity, however, we are simply talking about spontaneous symmetry breaking with respect to the symmetries related to the definition of time. Then the theory of massive gravity defines a preferred time-direction given by the Stückelberg function $T_0(r,t)$. In this

way, any observer defining the local time-coordinate $t \neq T_0(r,t)$ will perceive effectively a massive graviton. However, observers defining their local time as $t = T_0(r,t)$, will perceive effectively a mass-less graviton. We can conclude that the effective mass in the non-linear formulations of massive gravity is a relative concept since it depends on how the observers define their local time. Thus, a possible Higgs mechanism at the graviton level requires the introduction of new concepts like the definition of the relative effective mass for the graviton. The physics behind the dynamical origin of the graviton mass without any doubt affects the way we define the particle creation process of black-holes. In previous papers, it has been explained how the perception of particles depends on how the observers define their local time with respect to a preferred time-coordinate [10–12]. Then the same physics which reproduces a relative notion of particle due to the role of the graviton mass in the analysis of the black-hole radiation in massive gravity, explains the origin of the graviton mass dynamically. This is the case because the vacuum degeneracy is related to the preferred time-direction of the theory defined by the Stückelberg function ($T_0(r,t)$). Then if we define a positive frequency with respect to one vacuum, it might become negative when the same mode is perceived in a different vacuum [10–12]. The physics which we are considering here, suggests that any generator related to the time-direction in massive gravity, is potentially broken at the vacuum level. If we consider the decoupling limit of the theory [13–15], then the gauge transformations where the scalar degree of freedom is involved, are spontaneously broken since they are the ones related to the effective definition of the time-coordinate. Outside the decoupling limit, the consistency of the theory demands that the diffeomorphism transformations involving the three degrees of freedom, namely, scalar and vector components, are broken symmetries related to the existence of Nambu-Goldstone bosons. Again for this case, the three degrees of freedom are related to the way in which the time is defined locally. On the other hand, if we imagine a free-falling observer, then the Lorentz symmetry defined by he/she, is also spontaneously broken with respect to the boost symmetries which become the broken generators. From the perspective of the gauge/gravity duality, this is an interesting point because inside the AdS/CFT correspondence, the diffeomorphism invariance in the bulk, is equivalent to energy-momentum conservation on the boundary [16–20]. Then massive gravity, through the physics analyzed in this paper, offers an interesting scenario for the future research around the AdS/CFT correspondence. Note that at the decoupling limit of the theory, any diffeomorphism transformation involving the Stückelberg fields is of the form of a local $U(1)$ transformation in the complex plane [15]. In this special case, the observers defining the time in different directions perceive a set of vacuums, related to each other through rotations around the complex plane for the time coordinate. In massive gravity, we then have a special class of observers, defining the time in agreement with the Stückelberg function ($T_0(r,t)$) as has been remarked before. Note that $T_0(r,t)$ contains the information of the extra-degrees of freedom of the theory. This function has a trivial component which can be gauged away and a non-trivial component which contains the scales of the theory and the relevant information related to the extra-degrees of freedom. In this paper we demonstrate that the dynamical origin of the graviton mass, comes from the relation (or comparison) between the observers defining the time in agreement with $T_0(r,t)$ and the observers defining the time in a different direction ($t \neq T_0(r,t)$). Important physics related to the function $T_0(r,t)$ has been analyzed in [21,22]. We also demonstrate that the dynamical origin of the graviton mass is analogous to the standard Higgs mechanism for the scalar field for an ordinary $\sigma$-model. At the end of our analysis, our demonstration of the dynamical origin of the graviton mass shows that there is a fraction of the Christoffel connection which plays the role analogous to the gauge field in a $\sigma$-model since it contains terms depending explicitly on the Stückelberg function (preferred time-direction) at the perturbative level. This part does not vanish for a free-falling frame of reference. The other fraction of the connections corresponds to the usual GR contribution which vanishes for a free-falling frame of reference. The mathematical reason behind the non-vanishing of the connections depending explicitly on the function $T_0(r,t)$ is the fact that the dynamical metric in a free-falling frame of reference is not necessarily conformal to Minkowski, except when the Stückelberg function $T_0(r,t)$ corresponds trivially to the ordinary time-coordinate.

It is interesting to notice that the Stückelberg function has a double effect in this formulation: (1). It is the mass parameter which determines the location of the physical vacuum. (2). It reproduces the gauge portion of the connections, making them equivalent to gauge fields of the theory. The paper is organized as follows: In Section 2, we describe briefly the massive gravity formulation used in this paper. In Section 3, we explain briefly the Schwarzschild de-Sitter solution in dRGT massive gravity. In Section 4, we explain the standard Nambu-Goldstone theorem. In Section 5, we explain the standard Higgs mechanism for scalar fields. In Section 6, we analyze the massive action in a free-falling frame at the background level. We divide the analysis in different cases, depending on the relation between the free-parameters of the theory. In Section 7, we analyze the massive action, but this time at the perturbative level. In this case, the vacuum definition reveals a degeneracy and then the physical perturbations have to be expanded around the appropriate vacuum. In Section 8, we make a formal definition of the dynamical origin of the graviton mass. This part of the paper shows the analogy with respect to the standard Higgs mechanism for scalar fields. Finally, in Section 9, we conclude.

## 2. Massive Gravity Formulation

The coming analysis is general in the sense that it does not depend on which theory of massive gravity is used. However, for study purposes, we formulate the problem inside the dRGT formulation of massive gravity, which is well known inside the community. The action in the dRGT non-linear massive gravity formulation is defined as [9,13,14]

$$S = \frac{1}{2\kappa^2} \int d^4x \sqrt{-g}(R + m_g^2 U(g, \phi)).$$  (1)

This action contains three free-parameter, namely, the graviton mass, and the two parameters appearing inside the potential $U(g, \phi)$

$$U(g, \phi) = U_2 + \alpha_3 U_3 + \alpha_4 U_4,$$  (2)

where

$$U_2 = Q^2 - Q_2,$$  (3)

$$U_3 = Q^3 - 3QQ_2 + 2Q_3,$$  (4)

$$U_4 = Q^4 - 6Q^2 Q_2 + 8QQ_3 + 3Q_2^2 - 6Q_4,$$  (5)

$$Q = Q_1, \qquad Q_n = Tr(Q^n)^\mu{}_\nu,$$  (6)

$$Q^\mu{}_\nu = \delta^\mu{}_\nu - M^\mu{}_\nu,$$  (7)

$$(M^2)^\mu{}_\nu = g^{\mu\alpha} f_{\alpha\nu},$$  (8)

$$f_{\mu\nu} = \eta_{ab} \partial_\mu \phi^a \partial_\nu \phi^b.$$  (9)

We can then compute the field equations as follows

$$G_{\mu\nu} = -m^2 X_{\mu\nu},$$  (10)

where

$$X_{\mu\nu} = \frac{\delta U}{\delta g^{\mu\nu}} - \frac{1}{2} U g_{\mu\nu}.$$  (11)

Here $f_{\mu\nu}$ is the fiducial metric and $Q$ is the trace of the matrix $Q^\mu{}_\nu$.

## 3. The Schwarzschild de-Sitter Solution in dRGT

In [9], the S-dS solution was derived for two different cases. The first one corresponds to the family of solutions satisfying the condition $\beta = \alpha^2$, where $\beta$ and $\alpha$ correspond to the two free-parameters

inside the potential $U(g, \phi)$. In such a case, the Stückelberg function $T_0(r, t)$ becomes arbitrary. In other words, the vacuum solution is degenerate. There is a second family of solutions. They correspond to the case with two-free parameters satisfying the condition $\beta \leq \alpha^2$ with the Stückelberg function $T_0(r, t)$ constrained to some specific dependence on the space-time. These results reflect the fact that massive gravity behaves as a $\sigma$-model as has been remarked in [7,8]. Any solution satisfying the spherically symmetric condition, is expressed generically as

$$ds^2 = g_{tt}dt^2 + g_{rr}dr^2 + g_{rt}(drdt + dtdr) + r^2 d\Omega_2^2, \tag{12}$$

where

$$g_{tt} = -f(S_0 r)(\partial_t T_0(r, t))^2, \quad g_{rr} = -f(S_0 r)(\partial_r T_0(r, t))^2 + \frac{S_0^2}{f(S_0 r)}, \tag{13}$$

$$g_{tr} = -f(S_0 r)\partial_t T_0(r, t)\partial_r T_0(r, t),$$

with $f(S_0 r) = 1 - \frac{2GM}{S_0 r} - \frac{1}{3}\Lambda S_0^2 r^2$. Although the previous solution looks similar to the Schwarzschild-de Sitter metric in Einstein's gravity, it is not the same. In this case for example $T_0(r, t)$ is not an ordinary (time) coordinate, but it is rather a Stückelberg function which contains the information of the extra-degrees of freedom of the theory. We can visualize $T_0(r, t)$ as the part of the metric which generates the "weight" (mass) for the propagating degrees of freedom of the theory. The metric components in Equation (13), for example, are obtained after using the non-linear version of the Stückelberg trick expressed as

$$g_{\mu\nu} = \left(\frac{\partial Y^\alpha}{\partial x^\mu}\right)\left(\frac{\partial Y^\beta}{\partial x^\nu}\right) g'_{\alpha\beta}, \tag{14}$$

with the components of the Stückelberg function given by

$$Y^0(r, t) = T_0(r, t), \quad Y^r(r, t) = S_0 r. \tag{15}$$

In Equation (13), all the degrees of freedom of the theory are inside the dynamical metric. The fiducial metric in this case is just the Minkowskian one given explicitly as

$$f_{\mu\nu}dx^\mu dx^\nu = -dt^2 + dr^2 + r^2(d\theta^2 + r^2 sin^2\theta). \tag{16}$$

The dynamical metric when it contains all the degrees of freedom, is diffeomorphism invariant under the transformation [10–12,15]

$$g_{\mu\nu} \to \frac{\partial f^\alpha}{\partial x^\mu}\frac{\partial f^\beta}{\partial x^\nu}g_{\alpha\beta}(f(x)), \quad Y^\mu(x) \to f^{-1}(Y(x))^\mu. \tag{17}$$

In addition, the full action defined in Equation (1) is invariant under these previous sets of transformations.

## 4. The Nambu-Goldstone Theorem: The $\sigma$ Model

Here, as an example for the standard Nambu-Goldstone theorem and the subsequent Higgs mechanism, we consider the the linear $\sigma$-model. Consider this model for N scalar field $\phi(x)^i$. The Lagrangian is given by

$$\pounds = \frac{1}{2}(\partial_\mu \phi^i)^2 + \frac{1}{2}\mu^2(\phi^i)^2 - \frac{\lambda}{4}[(\phi^i)^2]^2, \tag{18}$$

which is invariant under the following transformation

$$\phi^i \to R^{ij}\phi^j. \tag{19}$$

This transformation is just a representation of the $O(N)$ group, namely, the group of orthogonal matrices in $N$ dimensions. The potential of the Lagrangian (18) is given by

$$V(\phi^i) = -\frac{1}{2}\mu^2(\phi^i)^2 + \frac{\lambda}{4}[(\phi^i)^2]^2. \tag{20}$$

This potential has a minimum when

$$(\phi_0^i)^2 = \frac{\mu^2}{\lambda}. \tag{21}$$

From this previous condition, we can determine the magnitude of $\phi_0^i$, but not its direction. Then the direction is in principle arbitrary. We can select some arbitrary direction, for example, we can select $\phi_0^i$ as

$$\phi_0^i = (0, 0, ..., 0, v), \tag{22}$$

with $v = \mu/\sqrt{\lambda}$. If we re-define the vacuum in agreement with the shift

$$\phi(x) = (\pi^k(x), v + \sigma(x)), \tag{23}$$

with $k = 1, ..., N - 1$. The Lagrangian in terms of the fields $\pi^k(x)$ ans $\sigma(x)$ becomes

$$\pounds = \frac{1}{2}(\partial_\mu\pi^k)^2 + \frac{1}{2}(\partial_\mu\sigma)^2 - \frac{1}{2}(2\mu^2)\sigma^2 - \sqrt{\lambda}\mu\sigma^3 - \sqrt{\lambda}\mu(\pi^k)^2\sigma - \frac{\lambda}{4}\sigma^4 - \frac{\lambda}{2}(\pi^k)^2\sigma^2 - \frac{\lambda}{4}[(\pi^k)^2]^2. \tag{24}$$

This Lagrangian clearly contains $N - 1$ massless $\pi^k$-fields and one massive field $\sigma$. Then there are $N - 1$ broken generators and they correspond to the Nambu-Goldstone bosons which move along the degenerate vacuum showed in the Figure 1.

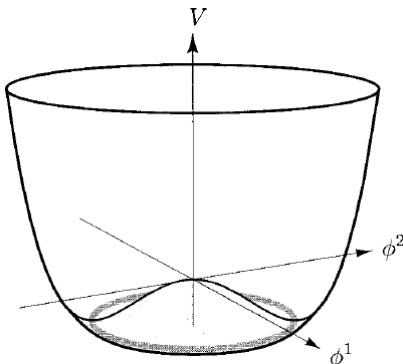

**Figure 1.** The potential for the spontaneous breaking of symmetry of the $O(N)$ symmetry, for the case of $N = 2$. Taken from [23].

### 4.1. The Nambu-Goldstone Theorem

Here we will summarize the results due to the Nambu-Goldstone theorem. We have seen in the previous explanations that the $\sigma$-model provides a basic example of the theorem. In this example, the symmetry $O(N)$ is broken and then the symmetry $O(N - 1)$ remains after selecting some vacuum. Here we will explain how the Nambu-Goldstone theorem can be applied to the standard linear $\sigma$-model. Note that in general, the potential $V(\phi^i)$, can be expanded as

$$V(\phi) = V(\phi_0) + \frac{1}{2}(\phi - \phi_0)^a(\phi - \phi_0)^b\left(\frac{\partial^2}{\partial\phi^a\partial\phi^b}V\right)_{\phi_0} + ..., \tag{25}$$

where $\partial V / \partial \phi^a = 0$, since we are expanding around a minimum. The mass matrix is symmetric and given by the coefficient

$$\left( \frac{\partial^2}{\partial \phi^a \partial \phi^b} V \right)_{\phi_0} = m_{ab}^2 \geq 0. \tag{26}$$

This previous condition is due to the fact that $\phi_0$ represents a minimum. At this point we will assume that the full action given by

$$\pounds = (kinetic \ terms) - V(\phi), \tag{27}$$

is invariant under the action of the group $G = O(N)$. In addition, we assume that the selected vacuum state is invariant under the action of a subgroup of $G$, given by $H = O(N-1)$. The vacuum state is not invariant under the action of the full group $G$. In summary, we have the following conditions

$$G: \quad \phi_0^{a'} = U(g) \phi_0^a \neq \phi_0^a, \tag{28}$$

$$H: \quad \phi_0^{a'} = U(h) \phi_0^a = \phi_0^a, \tag{29}$$

where $U(g)$ and $U(h)$ denote the representations of the groups $G$ and $H$ respectively. However, the potential $V(\phi)$ is still invariant under the action of the full group $G$. The action of this group on the potential expansion (25) gives the result

$$T^a(\phi) \frac{\partial}{\partial \phi^a} V(\phi) = 0, \tag{30}$$

where $T^a(\phi)$ is the generator of the group transformation. In the $U(g)$ representation, for example, it would take the form

$$U(g) = e^{T^a \alpha} \approx \hat{I} + \alpha T^a \rightarrow U(h) \phi_0^a = \phi_0^a + \alpha T^a(\phi_0), \tag{31}$$

where $T^a(\phi)$ denotes the action of the operator $T^a$ on the function state $\phi$. If we introduce the result (30) inside the expansion (25), then we get the condition

$$T^a(\phi_0) T^b(\phi_0) \frac{\partial^2}{\partial \phi^a \partial \phi^b} V(\phi) = T^a(\phi_0) T^b(\phi_0) m_{ab}^2 = 0. \tag{32}$$

Note that if $T^a(\phi_0) = 0$, namely, when the vacuum state selected is invariant under the action of the the group, then the corresponding mass component $m_{ab}$ is not necessarily zero. On the other hand, when the symmetry generator is broken, namely, when the group element belonging to $G$ does not leave the vacuum invariant, then $T^a(\phi_0) \neq 0$ and then the mass components related to this condition are necessarily zero ($m_{ab} = 0$). Then the number of broken generators are clearly related to the existence of gapless particles. In standard conditions, and for internal symmetries, the number of broken generators is equivalent to the number of Nambu-Goldstone bosons with linear dispersion relation [23]. However, under some special circumstances, it is possible to have more broken symmetries than Nambu-Goldstone bosons; in addition, under the same circumstances, the Nambu-Goldstone bosons have quadratic dispersion relations. An interesting analysis about this issue has been done in [24–33].

*4.2. Charge Conservation*

In quantum theory, the Goldstone theorem suggests that if there is a field operator $\phi(x)$ with non-vanishing vacuum expectation value $< 0|\phi(x)|0 >$ and in addition, the vacuum expectation value is not a singlet under the transformations of some representation of a symmetry group, then some

massless particles will appear in the spectrum states. It is well known that up to a total derivative term in the action (Lagrangian £), the conserved charge is given by

$$j_\mu^a(x) = \frac{\partial \pounds}{\partial(\partial^\mu \phi)} \frac{\delta \phi(x)}{\delta \alpha^a},$$ (33)

where $\delta \phi(x)/\delta \alpha$ corresponds to the field variations under symmetry transformations of the Lagrangian. The previously defined current is divergence-less and the corresponding charges are given by

$$Q^a(x) = \int d^3 x j_0^a(x).$$ (34)

In the standard cases, these charges are conserved, $dQ^a/dt = 0$, and they have a well defined commutation relations given by

$$[Q^a, Q^b] = C^{abc} Q^c,$$ (35)

where $C^{abc}$ are the structure constants of the Lie algebra. We can define an unitary operator with the charge being the generator of the group transformations

$$U = e^{iQ^a \alpha^a}.$$ (36)

If the vacuum is non-degenerate, then the previously defined charge annihilates the vacuum, namely, $U|0> = |0>$, or equivalently

$$Q^a|0> = 0.$$ (37)

When the vacuum is degenerate, then these previous conditions are not satisfied and in general

$$U \neq e^{iQ^a \alpha^a}, \quad Q^a|0> \neq 0.$$ (38)

This case is the interesting one for the purposes of this paper. It is not difficult to observe that in dRGT massive gravity, for the black-hole solutions, the quantity related to the translation under time-coordinates plays the role of the charge. In fact, the apparent non-conservation of the Komar and ADM mass (energy) in the non-liner theory of massive gravity, is a consequence of the fact that the vacuum is degenerate at scales where the extra-degrees of freedom of the theory become relevant [34,35]. The effect of the extra-degrees of freedom is to create distortion of the notions of time perceived by different observers located at scales where they become relevant. The distortions of time, specifically in its periodicity in the complex plane after analytical extension is what generates the effect of extra-particle creation process for a black-hole inside this theory [10–12]. As has been explained before, this extra-component of radiation comes from the mismatch between the periodicity of the function $T_0(r, t)$ and the standard time-coordinates $t$. These previous effects are all consequences of the fact that the time-like Killing vector does not point in the direction of the ordinary time-coordinate. Under these arguments, we can perceive once again a connection between the mechanism behind the dynamical origin of the graviton mass and the mechanism able to create deviations with respect to GR when we analyze the Black-hole evaporation process. If the operator $\phi(x)$ is not a singlet, then its commutation with the charge $Q^a$ is non-zero and given by

$$[Q^a, \phi'(x)] = \phi(x).$$ (39)

The vacuum expectation value of this operator is given by

$$< 0|Q^a \phi'(x) - \phi'(x) Q^a|0 > \neq 0.$$ (40)

If we introduce the definition of charge given in Equation (34), then we get

$$\sum_n \int d^3y \left[ <0|j_0^a(y)|n><n|\phi'(x)|0> - <0|\phi'(x)|n><n|j_0^a(y)|0> \right]_{x^0=y^0} \neq 0, \qquad (41)$$

where the equal-time condition has been imposed. After some trivial calculations, this previous expression finally becomes

$$(2\pi)^3 \sum_n \delta^{(3)}(\vec{p}_n) \left[ <0|j_0^a(0)|n><n|\phi'(x)|0> e^{iM_n y_0} - <0|\phi'(x)|n><n|j_0^a(0)|0> e^{-iM_n y^0} \right]_{x^0=y^0}, \qquad (42)$$

which must be different from zero in agreement with the previous results. Note that $p_n^0 = M_n$ and the spatial integrals were evaluated. The current conservation guarantees that the previous expression is independent of $y^0$. Then it is trivial to observe that $M_n = 0$ and this proves the Goldstone theorem. The proof elaborated in the previous way can be found in [36]. Note that the standard proof has to be modified when we analyze non-relativistic systems [30–33].

## 5. The Standard Higgs Mechanism: Local Gauge Symmetry

In the previous section we observed that the Nambu-Goldstone theorem is formulated by understanding the physics formulated in terms of group theory and the exact form of the potential is not important at the moment of understanding how many Nambu-Goldstone bosons appear after breaking the symmetry. Notice that in the previous section in order to get the Goldstone theorem result, it was enough to consider global symmetries for the Lagrangian. In this section, we will explain the Higgs mechanism. The Higgs mechanism appears when we consider local gauge symmetries. Consider for example the Lagrangian

$$\pounds = D_\mu \phi^* D^\mu \phi - m^2 \phi^* \phi - \lambda (\phi^* \phi)^2 - \frac{1}{4} F_{\mu\nu} F^{\mu\nu}, \qquad (43)$$

where $D_\mu = \partial_\mu + ieA_\mu$ is the corresponding covariant derivative, which transforms like the field $\phi$ under local gauge transformations. The potential for the previous Lagrangian is given by

$$V(\phi, \phi^*) = m^2 \phi^* \phi + \lambda (\phi^* \phi)^2. \qquad (44)$$

The action is invariant under the following transformations

$$\phi \to e^{-i\theta(x)}\phi, \quad \phi^* \to e^{i\theta(x)}\phi^*, \quad A^\mu \to A^\mu - \frac{1}{e}\partial_\mu \theta(x). \qquad (45)$$

The ground state is given by

$$|\phi| = \left( -\frac{m^2}{2\lambda} \right)^{1/2} = a. \qquad (46)$$

Note that $m$ cannot be interpreted at this point as the mass but rather as a parameter. In fact, $m^2 < 0$ is necessary in this case. Since the vacuum breaks the symmetry, this means that we are working around the wrong vacuum, then we can re-define it by using the following shift

$$\phi(x) = a + \frac{\phi_1(x) + i\phi_2(x)}{\sqrt{2}}. \qquad (47)$$

In terms of this field redefinition, the Lagrangian (43) becomes

$$\pounds = -\frac{1}{4}F^{\mu\nu}F_{\mu\nu} + e^2 a^2 A_\mu A^\mu + \frac{1}{2}(\partial_\mu \phi_1)^2 + \frac{1}{2}(\partial_\mu \phi_2)^2 - 2\lambda a^2 \phi_1^2 + \sqrt{2}ea A^\mu \partial_\mu \phi_2 + \dots \qquad (48)$$

Note that there is a mix between the kinetic term of the field $\phi_2$ and the massless vector field $A^\mu$. Then we can eliminate the $\phi_2$ field by doing the following transformations

$$\phi_1' = \phi_1 - \Lambda\phi_2,$$
$$\phi_2' = \phi_2 + \Lambda\phi_1 + \sqrt{2}\Lambda a. \tag{49}$$

After introducing these sets of transformations, we get

$$\pounds = -\frac{1}{4}F_{\mu\nu}F^{\mu\nu} + e^2a^2A_\mu A^\mu + \frac{1}{2}(\partial_\mu\phi_1)^2 - 2\lambda a^2\phi_1^2 + ... \tag{50}$$

We can observe that this Lagrangian has two fields, namely, one vector field $A^\mu$ which is massive and a second scalar field $\phi_1$, which is massless for a total of four degrees of freedom. The degrees of freedom (one) represented by the field $\phi_2$ has been eaten up by the vector field. Then the Nambu-Goldstone bosons produced after symmetry breaking, can be eaten up by other fields and then such fields become massive. The gauge represented by the Lagrangian (50) is called unitary gauge because it represents the gauge where physical particles will appear. The Higgs mechanism as explained in this section, is taken from [36].

## 6. Massive Gravity in a Free-Falling Frame Of Reference

Our analysis starts with the theory expanded around a free-falling frame of reference satisfying the condition $f(S_0 r) \to 1$ locally for any observer. The metric in a free-falling frame is not necessarily Minkowski inside the massive gravity formulation. Only when the Stückelberg function is trivially defined to be the standard time-coordinate $T_0(r,t) = S_0 t$, then the free-falling metric is conformal to Minkowski. The free-falling condition also constrains the behavior of the Stückelberg function for the case of two free-parameters for the massive action (without including the graviton mass), if we impose the regularity condition at the future event horizon as has been explained in [9]. For the case of one free-parameter with the Stückelberg function arbitrary, the free-falling condition does not provide any constraint on $T_0(r,t)$. The explicit form of the $Q$-matrix appearing inside the massive action will depend on the case under evaluation. Here we explore the two cases at the background level.

*6.1. Case (i). Two Free-Parameters Satisfying the Condition $\alpha^2 \geq \beta$*

In such a case, the Stückelberg function is given by [9]

$$T_0(r,t) = St \pm \int^{S_0 r}\left(\frac{1}{f(u)} - 1\right). \tag{51}$$

Note that in a free-falling frame, then $T_0'(r,t) = 0$ since $f(u) \to 1$. Then the $Q$-matrix given in Equations (7) and (8) becomes

$$Q^\mu_{\ \nu} = \left(1 - \frac{1}{S_0}\right)\hat{I}_{4\times 4}, \tag{52}$$

in agreement with the notation given in [9]. If we introduce this result inside Equations (3)–(5), then we obtain

$$U_2(g,\phi) = \frac{12(-1 + S_0)^2}{S_0^2}, \quad U_3(g,\phi) = \frac{24(-1 + S_0)^3}{S_0^3}, \quad U_4(g,\phi) = \frac{24(-1 + S_0)^4}{S_0^4}. \tag{53}$$

Note that for the case $S_0 = 1$ the massive action vanishes independent of the time orientation with respect to $T_0(r,t)$. This is consistent with what has been already found in [9]. On the other hand, for any value taken by the parameter $S_0$, at the background level, the massive action will behave as a cosmological constant term. A different case is expected at the perturbative level, where the massive action will have a non-trivial behavior (to be analyzed later). The previous results demonstrate that at

the background level and for a free-falling frame of reference, the symmetry under time-translations is not spontaneously broken. In other words, for this special case, there is no preferred time orientation and the physics will be independent on the frame of reference selected by the observers.

### 6.2. Case (ii). One Free-Parameter Case: Parameters Satisfying the Condition $\alpha^2 = \beta$

For the case of one free-parameter with the Stückelberg function arbitrary, the behavior is not as trivial as in the previous case. Here it is not possible to assume $T_0'(r, t) = 0$ for a free-falling frame, then the $Q$-matrix for this case becomes

$$
Q^{\mu}{}_{\nu} = \begin{pmatrix}
1 - \left( \dfrac{2 - \left( \frac{T_0'(r,t)}{S} \right)^2}{S \left( 4 - \left( \frac{T_0'(r,t)}{S} \right)^2 \right)^{1/2}} \right) & \dfrac{T_0'(r,t)}{S^2 \left( 4 - \left( \frac{T_0'(r,t)}{S} \right)^2 \right)} & 0 & 0 \\[2em]
- \dfrac{T_0'(r,t)}{S^2 \left( 4 - \left( \frac{T_0'(r,t)}{S} \right)^2 \right)} & 1 - \dfrac{2}{S \left( 4 - \left( \frac{T_0'(r,t)}{S} \right)^2 \right)^{1/2}} & 0 & 0 \\[2em]
0 & 0 & 1 - \frac{1}{S} & 0 \\[1em]
0 & 0 & 0 & 1 - \frac{1}{S}
\end{pmatrix}.
\tag{54}
$$

Note that if $T_0'(r, t) = 0$, we recover the previous case. Here if an observer defines the time in agreement with $T_0(r, t)$, such that he/she perceives the time as $T_0(r, t) = t$, then his/her $Q$-matrix will be defined in agreement with Equation (52). On the other hand, if the observer in a free-falling condition defines his/her time arbitrarily, such that $T_0(r.t) \neq t$, then his/her $Q$-matrix will be defined in agreement with Equation (54). In any case, even if these observers define the $Q$-matrix in this way, the terms related to $T_0(r, t)$ will not appear inside the massive action (54). Indeed, for this case we get

$$
U(g, \phi) = \frac{2 + 6\alpha(1 + \alpha)}{\alpha^4},
\tag{55}
$$

after introducing the result (54) inside Equations (3)–(5) and then using Equation (2). Note that here $\beta = \alpha^2$ and $S_0 = \alpha/(1 + \alpha)$. The result (55) is independent of $T_0(r, t)$. This means that the massive action again behaves as a standard cosmological constant term at the background level. Different situation will appear at the perturbative level. We can conclude at this point that at the background level in a free-falling frame of reference, the massive action operates as the standard cosmological constant in GR, independent of the relation between the two free-parameters of the theory. This result is evident because at this point we do not have degrees of freedom propagating through the vacuum. The degrees of freedom propagating through the vacuum will appear when we analyze the physics at the perturbative level.

## 7. Perturbation around a Free-Falling Frame

We will consider perturbations for the case where the observers are in a free-falling frame of reference. We will see that in these situations, the symmetries related to the time-translations are spontaneously broken for some specific combinations of the two free-parameters of the theory.

### 7.1. Case (i). Two Free-Parameters Satisfying the Condition $\alpha^2 \geq \beta$

In this case, the perturbation of the $Q$-matrices is done following the method used in [9]

$$
(\delta Q^{\mu}{}_{\sigma})(Q^{\sigma}{}_{\nu} - \delta^{\sigma}{}_{\nu}) + (Q^{\mu}{}_{\sigma} - \delta^{\mu}{}_{\sigma})\delta Q^{\sigma}{}_{\nu} = -\delta(M^2)^{\mu}{}_{\nu} = h^{\gamma}{}_{\beta}(M^2)^{\beta}{}_{\nu}.
\tag{56}
$$

Taking into account the result (52) at the background level and the generic perturbation Equations (56), we obtain the following results for the potential

$$\delta U(g,\phi) = \frac{\left[(1+2\alpha+\beta)^2(-1+2\alpha^3-2\alpha^2\sqrt{\alpha^2-\beta}+(-3+2\sqrt{\alpha^2-\beta})\beta-3\alpha(1+\beta))\right]h[r,t]}{(\alpha+\sqrt{\alpha^2-\beta}+\beta)^4}, \quad (57)$$

where $h(r,t)$ is the trace defined in agreement with the metric

$$ds^2 = S_0^2 ds_M^2, \quad (58)$$

with $ds_M^2$ being the standard Minkowski metric as in the Special Relativistic case. From this result, it is clear that the dynamical metric is trivially related to Minkowski through a conformal transformation through a constant conformal factor. The trace is explicitly given by $h(r,t) = (-h_{00}(r,t) + h_{rr}(r,t))/S_0^2$. With these previous results, we can expand the massive action $\sqrt{-g}U(g,\phi)$ up to second order. The perturbative expansion of the root square of the determinant for the dynamical metric is [37]

$$\sqrt{-g} \approx S_0^2\left(1+\frac{1}{2}h-\frac{1}{4}h^\alpha{}_\beta h^\beta{}_\alpha+\frac{1}{8}h^2\right), \quad (59)$$

then the massive action is clearly given by

$$\sqrt{-g}U(g,\phi) \approx S_0^2\left(1+\frac{1}{2}h-\frac{1}{4}h^\alpha{}_\beta h^\beta{}_\alpha+\frac{1}{8}h^2\right)(U(g,\phi)_{back}+\delta U(g,\phi)), \quad (60)$$

where $U(g,\phi)_{back}$ corresponds to the background value of the potential and $\delta U_{(g,\phi)}$ is the perturbation around the background. The relevant terms for the expansion up to second order in the graviton field are

$$\sqrt{-g}U(g,\phi) \approx S_0^2\left(1+\frac{1}{2}h-\frac{1}{4}h^\alpha{}_\beta h^\beta{}_\alpha+\frac{1}{8}h^2\right)U(g,\phi)_{back}+S_0^2\left(1+\frac{1}{2}h\right)\delta U(g,\phi). \quad (61)$$

Here the first term on the right-hand side behaves as a cosmological constant one expanded up to second order in perturbation theory. For simplicity, we can express the perturbation of the potential $U(g,\phi)$ as $\delta U(g,\phi) = F(\alpha,\beta)h$, with $\alpha$ and $\beta$ representing the two-free parameters in addition to the graviton mass. Here $F(\alpha,\beta)$ is defined in agreement with Equation (57). Note that $S_0$ is a function of the same set of parameters. Then Equation (61), is equivalent to

$$\sqrt{-g}U(g,\phi) \approx S_0^2 U(g,\phi)_{back}+h\left(\frac{S_0^2}{2}U(g,\phi)_{back}+S_0^2 F(\alpha,\beta)\right)-\frac{S_0^2}{4}h^\alpha{}_\beta h^\beta{}_\alpha U(g,\phi)_{back}$$

$$+h^2\left(\frac{S_0^2}{8}U(g,\phi)_{back}+\frac{S_0^2}{2}F(\alpha,\beta)\right). \quad (62)$$

If we define the potential in agreement with $\sqrt{-g}U(g,\phi) = V(g,\phi)$, then we can find the vacuum solutions if we solve

$$\frac{dV(g,\phi)}{dh_{\mu\nu}} = 0, \quad (63)$$

which in this case is equivalent to

$$S_0^4\eta_{\mu\nu}\left(\frac{1}{2}U(g,\phi)_{back}+F(\alpha,\beta)\right)-\frac{S_0^2}{2}U(g,\phi)_{back}h_{\mu\nu vac}+\eta_{\mu\nu}\left(\frac{1}{4}U(g,\phi)_{back}+F(\alpha,\beta)\right)h_{vac}=0, \quad (64)$$

where $\eta_{\mu\nu}$ is just the standard Minkowski metric which is conformally related to the real metric (see Equation (58)) and the subindex *vac* makes reference to the vacuum solutions. Solving for this previous expression, we get

$$h_{00vac} = -h_{rrvac} = A(\alpha, \beta), \qquad h_{0rvac} = 0, \tag{65}$$

where $A(\alpha, \beta)$ is a function of $\alpha$ and $\beta$. Figure 2 shows the behavior for each vacuum component. Note that for the case $\alpha = 0$ or $\beta = 0$, the vacuum field vanishes. This behavior can be better visualized in Figure 3. We can notice that even if the vacuum field does not vanish for different values of the parameters satisfying the condition $\alpha^2 \geq \beta$, the Stückelberg function $T_0(r, t)$ does not appear for this case. This means that we have a unique vacuum for a fixed combination of the parameters $\alpha$ and $\beta$. Then the symmetry is not spontaneously broken and the observers cannot perceive Nambu-Goldstone bosons and there is no way to see any dynamical origin of the graviton mass at this level. For example, here we can evaluate the mass matrix for the present potential by obtaining the second derivatives with respect to graviton fields in Equation (62). The parameter-dependent matrix mass is given by

$$m_{ab}^2 = -\frac{S_0^2}{2} U_{back} \delta^\mu{}_\alpha \delta^\nu{}_\beta + S_0^2 \eta_{\mu\nu} \eta_{\alpha\beta} \left( \frac{1}{4} U_{back} + F(\alpha, \beta) \right), \tag{66}$$

where the subindex *ab* represents the matrix components after doing the tensorial product in this previous expression. From this result, we can find the eigenvalues, which would correspond the parameter-dependent mass for some specific modes. Here, however, the eigenvalues would not correspond to a dynamical graviton mass because there is no gauge-field associated to the masses and the Stückelberg function is absent as has just been mentioned. This means that for this case, even at the perturbative level, the massive action behaves as an ordinary cosmological constant (unique vacuum).

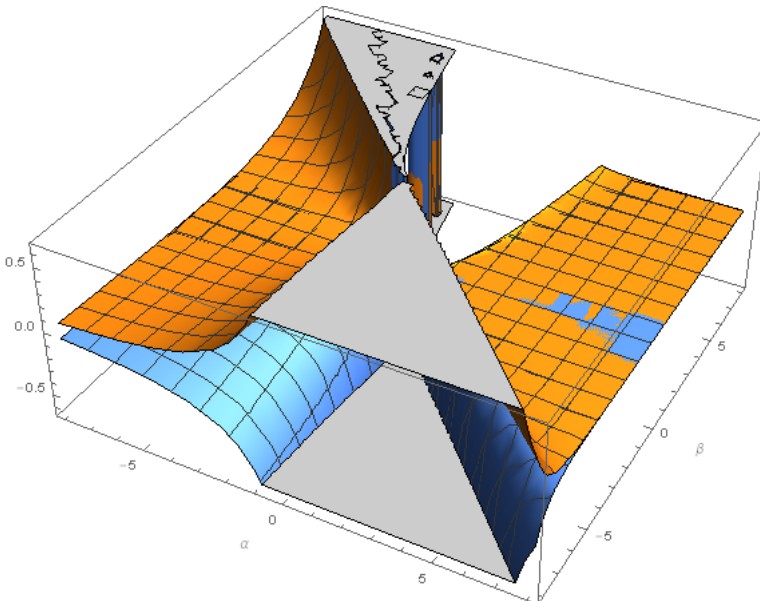

**Figure 2.** The vacuum state representative for the potential (62) as a function of the parameters $\alpha$ and $\beta$. The yellow color plot corresponds to the vacuum component for $h_{00}$ and the blue color corresponds to the component $h_{rr}$. It is easy to visualize the symmetry between both components.

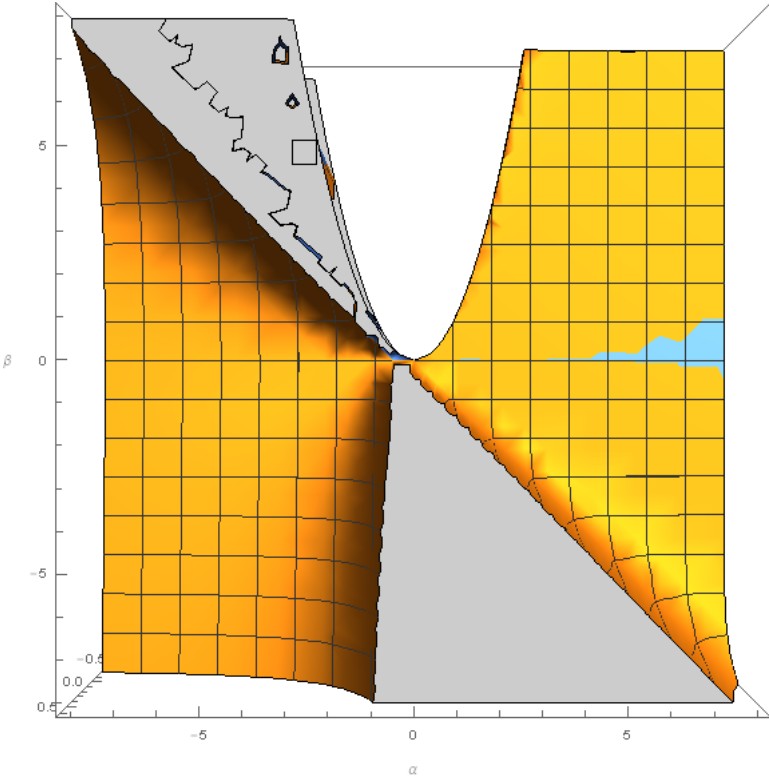

**Figure 3.** The plane $\alpha$-$\beta$ corresponding to the vacuum state of Figure 2. Here we can perceive a parabolic behavior between both parameters in some regions of the plane. This portion of the figure corresponds to the vacuum state for $h_{00}$. The vacuum state for $h_{rr}$ is the mirror image of this figure.

### 7.2. Case (ii). One Free-Parameter Case: Parameters Satisfying the Condition $\alpha^2 = \beta$

For this case, the condition $T'_0(r, t) = 0$ is not necessarily satisfied for a free-falling reference frame and then the structure of the perturbation for the $Q$-matrices is not as trivial as in the previous case. By using the same procedures based in the generic formulation developed in [9], then we can divide the analysis in two cases.

#### 7.2.1. Vanishing Spatial Dependence of the Stückelberg Function: $T'_0(r, t) = 0$

More generically, this case would correspond to the situations where $dT_0(r, t) = S_0$. This is the case because in general, $T_0(r, t) = S_0 t + A(r, t)$. Then

$$dT_0(r, t) = \dot{T}_0(r, t)dt + T'_0(r, t)dr = S_0 dt + \left( \dot{A}(r, t)dt + A'(r, t)dr \right). \tag{67}$$

Then if $T_0(r, t)$ is supposed to represent the ordinary time-coordinate, the condition $\dot{A}(r, t)dt + A'(r, t)dr = 0$ has to be satisfied. This does not necessarily mean $T'_0(r, t) = A'(r, t) = 0$; however, for the sake of simplicity, in this section we will work under such special condition ($T'_0(r, t) = 0$), since we will take $\dot{A}(r, t) = 0$ which is equivalent to the stationary condition assumption. Note that the present case is reduced to the one with two free-parameters if we just take into account the condition $\alpha = \beta^2$ in Equation (62). It is trivial to demonstrate that

$$\sqrt{-g}U(g, \phi)_{back} \approx \frac{2 + 6\alpha(1 + \alpha)}{\alpha^2(1 + \alpha)^2}, \tag{68}$$

and

$$\sqrt{-g}\delta U(g, \phi) \approx -\frac{(1 + \alpha)}{\alpha^2}h(r, t). \tag{69}$$

Then the total action, expanded up to second order in perturbation theory, is defined with the help of the generic result (61) as

$$\sqrt{-g}U(g,\phi) \approx \left(1 + \frac{1}{2}h - \frac{1}{4}h^{\alpha}{}_{\beta}h^{\beta}{}_{\alpha} + \frac{1}{8}h^2\right)\left(\frac{2 + 6\alpha(1+\alpha)}{\alpha^2(1+\alpha)^2}\right) - \left(1 + \frac{1}{2}h\right)\left(\frac{1+\alpha}{\alpha^2}\right)h. \tag{70}$$

In this case, it can be demonstrated that it is impossible to find a vanishing coefficient for the trace $h(r,t)$ and at the same time obtain the Fierz-Pauli tuning, unless we accept divergent values. In fact, if $\alpha = 0$, then the linear term in the trace is absent and the Fierz-Pauli tuning appears but the graviton becomes strongly coupled and its massive modes cannot propagate. If we evaluate the condition (63), then we obtain for the potential (70) the following result

$$\bar{\eta}_{\mu\nu}\left[\frac{1}{2}U_{back} + \frac{1}{4}U_{back}h - \frac{1}{2}\left(\frac{1+\alpha}{\alpha^2}\right)h - \left(1 + \frac{1}{2}h\right)\left(\frac{1+\alpha}{\alpha^2}\right)\right] - \frac{1}{2}h_{\mu\nu}U_{back} = 0, \tag{71}$$

where $\bar{\eta}_{\mu\nu}$ is the background metric which is trivially related to Minkowski through the conformal transformation due to the factor $S_0$ as in Equation (58). Solving for the graviton field, we find that the vacuum is represented by the following set of solutions

$$h_{00vac} = -\frac{(\alpha^2 + 3\alpha^3 + 2\alpha^4 - 2\alpha^5 - \alpha^6)}{2(1+\alpha)^5}, \quad h_{rrvac} = \frac{(\alpha^2 + 3\alpha^3 + 2\alpha^4 - 2\alpha^5 - \alpha^6)}{2(1+\alpha)^5},$$

$$h_{0rvac} = 0, \tag{72}$$

where the subindex *vac* makes reference to the vacuum state. From Figure 4, we can observe the behavior for the vacuum state as a function of the free-parameter $\alpha$. Note that in general the trace of graviton field is non-zero, except for two well defined values of the parameter $\alpha$. Although the plot covers a large range of values, it is understood that the condition $|h| << 1$ must be satisfied. Note that the vacuum satisfies the relation $h_{00}/h_{rr} = -1$. Then the vacuum state represented by the potential (70) is given by a straight line with negative slope, where the axes are represented by the components $h_{00}$ and $h_{rr}$ as it is illustrated in Figure 5. The determinant for the vacuum solution given in eq. (72) can be seen graphically in the Figure 6. The same logic presented here applies to the previously analyzed case. If we compute the matrix of second derivatives for the potential (70), then we will discover that the eigenvalues for the mass matrix is parameter-dependent and equivalent to

$$\frac{\partial^2 V(g,\phi)}{\partial h_{\mu\nu}\partial h_{\alpha\beta}} = \left(\frac{1}{4}V_{back} - \left(\frac{1}{1+\alpha}\right)\right)\eta_{\mu\nu}\eta_{\alpha\beta} - \frac{1}{2}V_{back}\delta^{\mu}{}_{\alpha}\delta^{\nu}{}_{\beta}. \tag{73}$$

Compare this result with Equation (66). This matrix only represents two independent equations after developing the tensorial product. The result (73) represents the graviton masses for the different modes. However, again we have a single vacuum state (once we fix the parameters) since there is no path of the extra-degrees of freedom (Nambu-Goldstone bosons) appearing through a non-trivial function $T_0(r,t)$. In other words, there is no gauge field associated to the graviton mass in this case and since the vacuum is unique, in this special situation we do not have any dynamical origin for the graviton mass. There are two different eigenvalues for the matrix Equation (73). They are represented in Figures 7 and 8. Note that both eigenvalues can be zero for some specific values taken by the parameter $\alpha$.

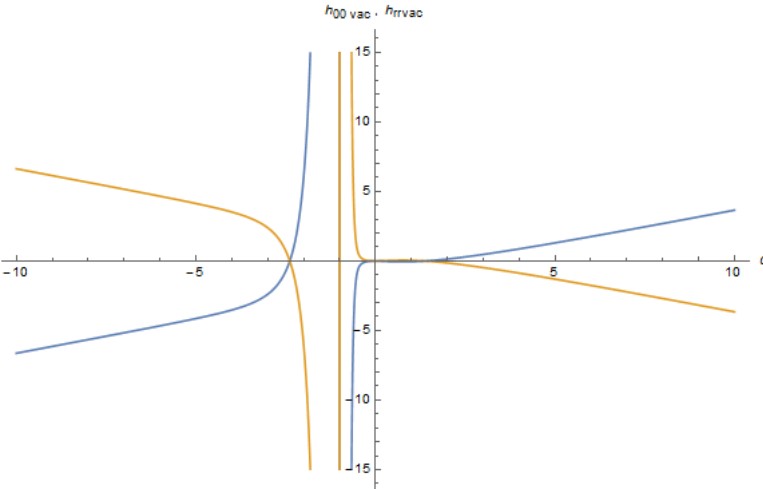

**Figure 4.** The vacuum state representative for the potential (70) as a function of the free-parameter $\alpha$. The blue line corresponds to the component $h_{00vac}$ of the graviton field. The yellow line corresponds to the component $h_{rrvac}$. Note that in general the trace is non-zero. It becomes zero for the case ($h_{00vac} = 0$ and $h_{rrvac} = 0$)obtained for two different values of the parameter $\alpha$. The condition $|h| << 1$ is assumed even if in the figure this constraint does not appear explicitly.

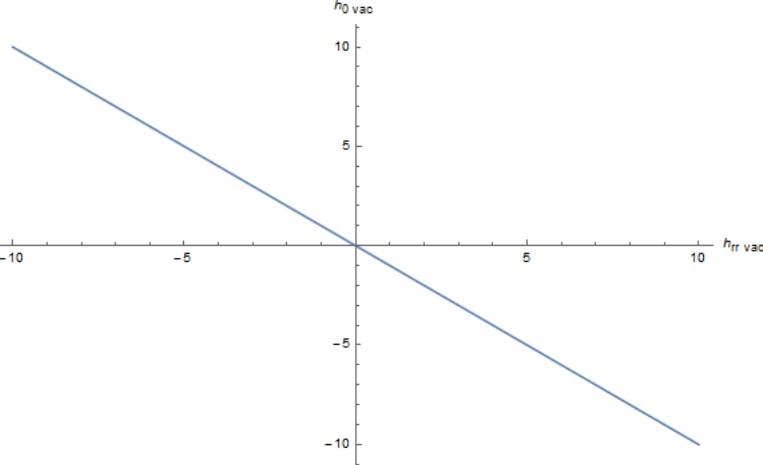

**Figure 5.** The same vacuum state represented by Figure 4. Note that there is a fixed relation between the vacuum components $h_{00vac}(r,t)$ and $h_{rrvac}(r,t)$. All the line represents the possible vacuum states depending on the parameter $\alpha$.

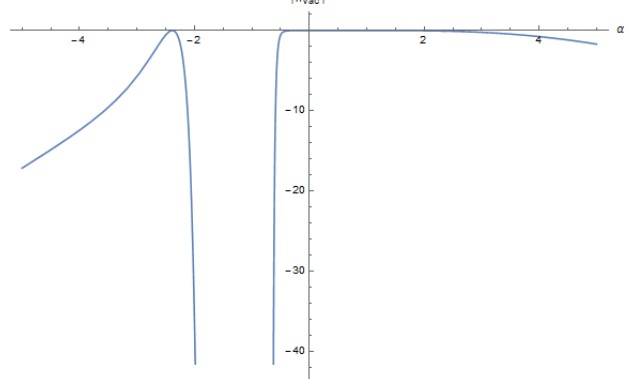

**Figure 6.** The determinant of the vacuum matrix corresponding to the graviton field $h$ and the solutions (72). Note that the determinant vanishes for some specific values of $\alpha$, consistent with the vanishing values of $h_{00vac}$ and $h_{rrvac}$.

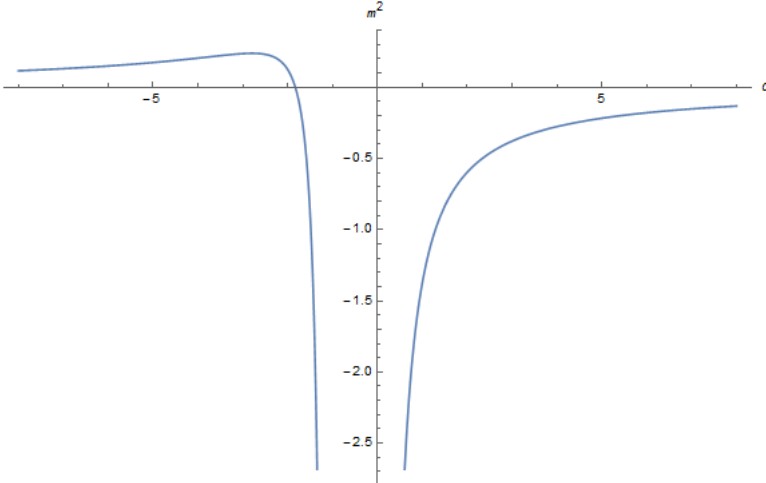

**Figure 7.** The parametric behavior of one of the eigenvalues for the matrix Equation (73). We can observe that the parametric mass becomes zero for some specific value of $\alpha$ for this case.

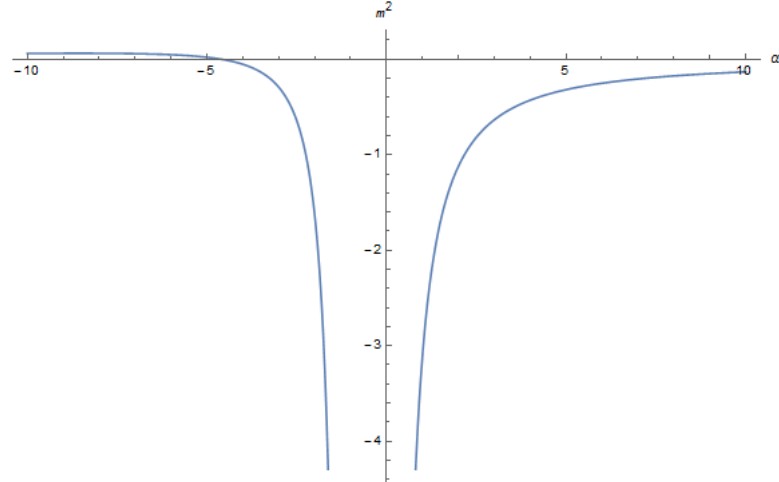

**Figure 8.** The parametric behavior of one of the eigenvalues for the matrix Equation (73). This case corresponds to a different eigenvalue with respect to the case illustrated in Figure 7. We can observe that the parametric mass becomes zero for some specific value of $\alpha$ for this case as in the previous case.

### 7.2.2. Non-Vanishing Spatial Dependence of the Stückelberg Function: $T_0'(r,t) \neq 0$

More generically, this case corresponds to the one where $dT_0(r,t) \neq S_0$, since $\dot{A}(r,t)dt + A'(r,t)dr \neq 0$. Here once again for simplicity we will take $\dot{A}(r,t) = 0$. However, there is no loss of generality in the arguments of this section by using this assumption. The extension to the general case is direct. The condition $\dot{A}(r,t = 0$ is equivalent to the stationary condition for the dynamical metric. This is the case because the stationary condition suggests that $\dot{T}_0(r,t) = S_0$, with $S_0$ being defined such that the dynamical metric is conformally trivial when the spatial dependence of the Stückelberg function vanishes and in addition, gravity effects are absent at all (free-falling condition). In this small section, we take $T_0'(r,t) \neq 0$. This is possible given the arbitrariness of the Stückelberg function for this case. Then the symmetry is now spontaneously broken and we do not have a single vacuum.

This situation is the most interesting for this paper. The potential expanded up to second order in perturbations becomes

$$\sqrt{-g}U(g,\phi) \approx \left(1 + \frac{1}{2}h - \frac{1}{4}h^\alpha{}_\beta h^\beta{}_\alpha + \frac{1}{8}h^2\right)\left(\frac{2 + 6\alpha(1+\alpha)}{\alpha^2(1+\alpha)^2}\right)$$

$$-\left(1 + \frac{1}{2}h\right)\left(-\frac{(1+\alpha)}{\alpha^2}h + \frac{2T_0'(r,t)(1+\alpha)^4}{\alpha^5}h_{0r} - \frac{T_0'(r,t)^2(1+\alpha)^5}{\alpha^6}\right) + ..., \tag{74}$$

We will consider that $T_0'(r,t) \approx M << 1$ in order to include the effect of the extra-degrees of freedom through perturbation. In other words, we only consider infinitesimal departures from the trivial result $T_0'(r,t) = 0$. Then here we consider $T_0'(r,t)^2$ as a second order term in the expansion of the action.

Here we can also calculate the vacuum condition by using Equation (63). The result is

$$\bar\eta_{\mu\nu}\left[\frac{1}{2}V(g,\phi)_{back} + \frac{1}{4}V_{back}h + \frac{1+\alpha}{2\alpha^2}h - \frac{T_0'(r,t)(1+\alpha)^4}{\alpha^5}h_{0r} + \left(1 + \frac{1}{2}h\right)\left(\frac{1+\alpha}{\alpha^2}\right)\right]$$

$$+\bar\eta_{\mu\nu}\left(\frac{T_0'(r,t)^2(1+\alpha)^5}{2\alpha^6}\right) - \frac{1}{2}h_{\mu\nu}V_{back} - \left(1 + \frac{1}{2}h\right)\left(\frac{2T_0'(r,t)(1+\alpha)^4}{\alpha^5}\right)\delta^0{}_\mu\delta^r{}_\nu = 0, \tag{75}$$

where $\bar\eta_{\mu\nu}$ corresponds to the dynamical metric in the absence of gravity. However, in this case, due to the presence of the Stückelberg function, the metric is not conformal to Minkowski under the free-falling condition. In this case, it is given by

$$ds^2 = \bar\eta_{\mu\nu}dx^\mu dx^\nu = S_0^2\left(-dt^2 + dr^2\left[1 - \left(\frac{T_0'(r,t)}{S_0}\right)^2\right] - 2\frac{T_0'(r,t)}{S_0}dtdr + r^2d\Omega^2\right). \tag{76}$$

Note that for the case $T_0'(r,t) = 0$, we recover the previous results. By replacing the metric (76) inside Equation (75) and then solving the system of equations involved, we obtain

$$h_{oovac} = \frac{\alpha^2[2 + \alpha(6 + \alpha(6+\alpha))]}{2(1+\alpha)^5} + 2T_0'(r,t)^2 A(\alpha),$$

$$h_{rrvac} = -\frac{\alpha^2[2 + \alpha(6 + \alpha(6+\alpha))]}{2(1+\alpha)^5} + 2T_0'(r,t)^2 B(\alpha), \tag{77}$$

$$h_{0rvac} = \frac{T_0'(r,t)[-2(1+\alpha)^7 + \alpha[1 + 3\alpha(1+\alpha)][2 + \alpha(6 + \alpha(6+\alpha))]]}{2(1+\alpha)^4(1 + 3\alpha(1+\alpha))},$$

where $A(\alpha)$ and $B(\alpha)$ are two functions depending on the parameter $\alpha$. Note that the corrections due to the Stückelberg function presence are quadratic at the lowest order for $h_{oovac}$ and $h_{rrvac}$. They are also of linear order at the lowest order for $h_{0rvac}$. If we work for an infinitesimal value of $T_0'(r,t)$, then the relation between the vacuum components $h_{oovac}$ and $h_{rrvac}$ would be exactly of the same form as in Figures 4 and 5. Only the higher order contributions of the Stückelberg function break this mirror symmetry between these two components. The non-diagonal component $h_{0rvac}$ is linear in $T_0'(r,t)$ and it does not exist for a vanishing value of the Stückelberg function. The Figures 9 and 10 represent the behavior of the vacuum solutions (77). Note that in the region around $T_0'(r,t) \to 0$ ($M$ in the figure), there is a multiplicity of values taken by $\alpha$, for which we have absence of gravitons at the vacuum level.

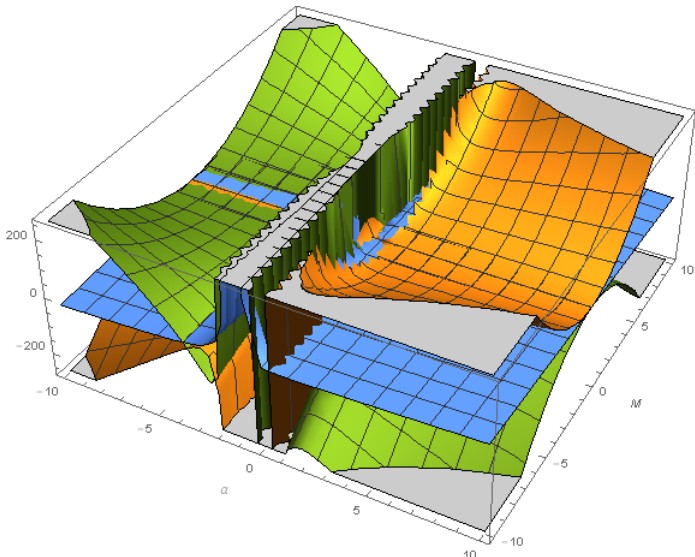

**Figure 9.** The behavior of the vacuum when the Stückelberg function is non-zero for the solutions (77). The components $h_{00}$ (yellow color) and $h_{rr}$ (green color) are mirror images of each other. The blue color corresponds to the vacuum solution for the $h_{0r}$. Here the Stückelberg function is parametrized in agreement with $M$. The Stückelberg function has been expanded up to second order.

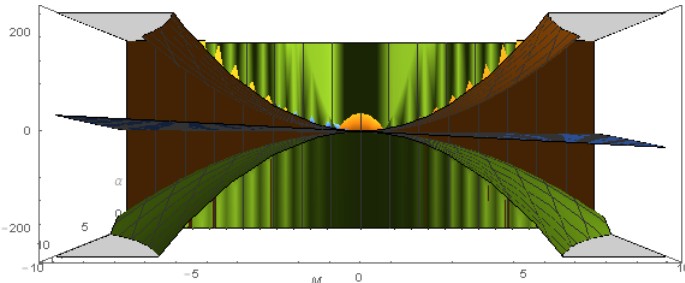

**Figure 10.** The same Figure 9 but observed from a different angle. From this perspective, the symmetry between the $h_{00}$ and $h_{rr}$ components of the graviton field in its vacuum state is clear. The symmetry disappears at the quadratic order in the Stückelberg function expansion.

We can calculate the determinant for the vacuum solution (77). It becomes a complicated solution in terms of $\alpha$ and $T_0'(r,t)$. Here we express the result as

$$|h_{\mu\nu}|_{vac} = C(\alpha) + T_0'(r,t)^2 D(\alpha) = v. \tag{78}$$

The vacuum is then degenerate and the behavior of this determinant can be observed from the Figure 11. The constant $C(\alpha)$ represents the trivial part representing the ordinary cosmological constant. The non-trivial contribution comes from the Stückelberg function $T_0'(r,t)$, which makes the vacuum degenerate.

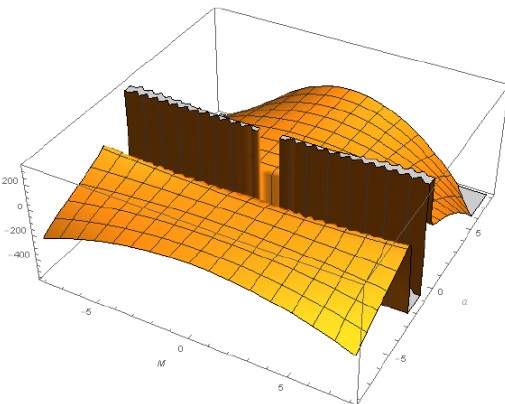

**Figure 11.** The determinant of the vacuum solution given in Equation (77). Note that around the region $T_0'(r,t) \to 0$, there exists an extremal condition for the value of the determinant. On the other hand, other extremal conditions appears with respect to $\alpha$.

An interesting value, is the one for which the determinant vanishes. In fact, $v = 0$ in Equation (78) when

$$T_0'(r,t) = \pm \left( -\frac{C(\alpha)}{D(\alpha)} \right)^{1/2}. \tag{79}$$

Here we can also calculate the eigenvalues for the mass matrix given by the second derivative of the potential (74). However, what is interesting to notice at this point is that the dynamical mass of the graviton comes from the spatial dependence of the Stückelberg function $T_0'(r,t)$. In the next section, we will make a formal definition of the dynamical origin of the graviton mass. We will see that $T_0'(r,t)$ has a double role, namely, not only does it appear as the mass parameter of the theory, but in addition, it provides the gauge-part for the Christoffel connections to behave as gauge-fields able to provide the graviton mass dynamically.

## 8. Symmetry Breaking Pattern and the Dynamical Origin of the Graviton Mass

From the previous analysis, it is clear that there is a degeneracy in the vacuum solution and that the Stückelberg function has two roles. It is related to the mass parameter of the theory, analogous to the $\mu$-parameter in the case of a scalar field (see Equation (21)). This can be observed explicitly for the case where the condition $\beta = \alpha^2$ is satisfied. The mass parameter determines the position of the false vacuum of the theory. In general, the action is invariant under the full dipheomorphism transformation defined as in Equation (17). The vacuum represented by Equation (78) is in general non-zero and then any physical perturbation has to be expanded around it. The Stückelberg function behaves as a preferred time direction for the vacuum field. In this case, its variation $dT_0(r,t)$ represents the order parameter of the theory. If an observer in a free-falling defines the Lorentz symmetry as $SO(3,1)$ ($G$), then after selecting some specific frame of reference, the preferred time-direction at the vacuum level will break the symmetry toward an ordinary rotation $SO(3)$ ($H$). The remaining $U(1)$ transformations related to the time-coordinate, correspond to the different frames of reference taken by the observers. They define the coset $G/H$ space. Due to the preferred time-direction $T_0(r,t)$, the observers defining the time in agreement with this Stückelberg function will not perceive the dynamical part of the graviton mass. This means that they will not perceive locally the spatial (temporal) variations of the Stückelberg function. At this level, any symmetry transformation involving the time should be considered as a potential broken generator depending on how the observers define their notion of time (frame of reference). If the symmetry under consideration is the Lorentz one, then we have three potential broken generators corresponding to the three boost directions. This is precisely the number of Nambu-Goldstone bosons that massive gravity theories contain (one degree of freedom for the

scalar and other two for vectors). When the observers select their frame of reference, the symmetry breaking pattern is given by

$$SO(3,1) \quad (G) \rightarrow SO(3) \quad (H), \quad G/H = U(1). \tag{80}$$

Here the generators of the $SO(3)$ symmetry are the different components of the angular momentum. The previous expression then suggests that the symmetry under spatial rotations is still valid after the observers make the selection of frame of reference. In other words, the angular momentum is still well defined after the symmetry is broken. The symmetry breaking patter illustrated in Equation (80) explains why in [10–12], it was discovered that under the spherically symmetric assumption, as far as the time coordinate is excluded, everything in dRGT massive gravity is exactly the same as in the GR case. Only when dynamical processes are considered, the departures with respect to GR are evident. In the Hawking radiation analysis, for example, from the path integral formulation analyzed in [10–12], it is evident that an extra-component or radiation will appear due to the distortions of the periodicity patterns of the propagators produced by the extra-degrees of freedom. The periodicity of the propagator is related to the $U(1)$ symmetry, which corresponds to the group of broken generators (coset space). Then the results of the present manuscript can also be perceived as a connection between the usual spontaneous symmetry breaking mechanism and the conformal symmetry breaking mechanism which is the responsible for the Hawking radiation effect. If the vacuum values for the graviton field are non-zero as has been found in Equations (77) and in the vacuum determinant given by Equation (78), then the physical perturbations have to be expanded around this degenerate vacuum. By repeating the standard techniques, we can shift the solution such that we work around the real vacuum. Then we define

$$h'_{\mu\nu} = h_{\mu\nu} + h_{\mu\nu vac}, \tag{81}$$

where $h_{\mu\nu vac}$ is defined in agreement with the vacuum components (77). Then the action has to take the corresponding shift. If we take the Riemmann tensor to be

$$R_{\lambda\mu\nu\kappa} = \frac{1}{2} \left( \frac{\partial^2 g_{\lambda\nu}}{\partial x^\kappa \partial x^\mu} - \frac{\partial^2 g_{\mu\nu}}{\partial x^\kappa \partial x^\lambda} - \frac{\partial^2 g_{\lambda\kappa}}{\partial x g_{\lambda\nu} x^\mu} + \frac{\partial^2 g_{\mu\kappa}}{\partial x^\nu \partial x^\lambda} \right) + g_{\eta\sigma} \left( \Gamma^\eta_{\nu\lambda} \Gamma^\sigma_{\mu\kappa} - \Gamma^\eta_{\kappa\lambda} \Gamma^\sigma_{\mu\nu} \right). \tag{82}$$

With the appropriate contractions, we can then find the Ricci scalar as $g^{\lambda\nu} g^{\mu\kappa} R_{\lambda\mu\nu\kappa}$. At the linear level, if we want to compute the kinetic term for the action, we can ignore the power expansion for $\sqrt{-g}$ in Equation (1). This expansion, however, cannot be ignored when we consider the potential (massive action). At the background level, then we can use $\sqrt{-g} = S_0^2$. By taking into account the dynamical metric components given in Equation (76), some terms of the form

$$\frac{\partial^2}{\partial x^\kappa \partial x^\mu} g_{rr} = \frac{\partial^2}{\partial x^\kappa \partial x^\mu} h_{rr} - 2(T_0''(r,t))^2 \delta^\mu_r \delta^\kappa_r - 2T_0'(r,t) T_0'''(r,t) \delta^\mu_r \delta^\kappa_r,$$

$$\frac{\partial^2}{\partial x^\kappa \partial x^\mu} g_{0r} = \frac{\partial^2}{\partial x^\kappa \partial x^\mu} h_{0r} - S_0 T_0'''(r,t) \delta^\mu_r \delta^\kappa_r, \tag{83}$$

will appear at the end of the calculations for the first term on the right hand-side of Equation (82). Note that in Equation (83) we have not yet introduced the physical perturbation (81). However, this is a trivial part and it will only introduce new terms depending on the derivatives of $T_0(r,t)$. We only have to take into account that in general

$$\frac{\partial^2}{\partial x^\kappa \partial x^\beta} h_{\mu\nu} = \frac{\partial^2}{\partial x^\kappa \partial x^\beta} h'_{\mu\nu} + \delta^\kappa_r \delta^\beta_r \frac{\partial^2}{\partial x^\kappa \partial x^\beta} h_{\mu\nu vac}. \tag{84}$$

Then we can easily calculate the derivatives around the physical vacuum defined in Equation (77) for the different components if we use the results

$$\frac{\partial^2}{\partial x^\kappa \partial x^\beta} h_{00vac} = A(\alpha)[4(T_0''(r,t))^2 + 4T_0'(r,t)T_0'''(r,t)]\delta^\kappa{}_r \delta^\beta{}_r,$$

$$\frac{\partial^2}{\partial x^\kappa \partial x^\mu} h_{rrvac} = B(\alpha)[4(T_0''(r,t))^2 + 4T_0'(r,t)T_0'''(r,t)]\delta^\mu{}_r \delta^\kappa{}_r,$$

$$\frac{\partial^2}{\partial x^\kappa \partial x^\mu} h_{0rvac} = \frac{T_0'''(r,t)[-2(1+\alpha)^7 + \alpha[1+3\alpha(1+\alpha)][2+\alpha(6+\alpha(6+\alpha))]]}{2(1+\alpha)^4(1+3\alpha(1+\alpha))}. \tag{85}$$

For these previous terms, all the other expressions are standard. Regarding the linear-order approximation for the second term for the Riemann tensor, we have to calculate the connections as follows

$$\Gamma^\lambda_{\mu\nu extra1} = \frac{1}{2}\frac{T_0'(r,t)^2}{S_0^4}\delta^\lambda{}_0 \delta^\rho{}_0 \left(\frac{\partial}{\partial x^\nu} g_{\rho\mu}\right)$$

$$\Gamma^\lambda_{\mu\nu extra2} = -\frac{1}{2}\frac{T_0'(r,t)}{S_0^3}\delta^\lambda{}_0 \delta^\rho{}_r \left(\frac{\partial}{\partial x^\nu} h_{\rho\mu}\right)$$

$$\Gamma^\lambda_{\mu\nu extra3} = \frac{1}{2}\frac{\eta^{\lambda\rho}}{S_0^2}\left(\frac{\partial}{\partial x^\nu} h_{\rho\mu} + 2\frac{T_0'(r,t)T_0''(r,t)}{S_0^4}\delta^r{}_\nu \delta^0{}_\rho \delta^0{}_\mu - \frac{T_0''(r,t)}{S_0^3}\delta^r{}_\nu \delta^0{}_\rho \delta^r{}_\mu\right), \tag{86}$$

with all the other terms standard. Again in this case, we still have to expand the graviton field around the physical vacuum, such that we can work with the physical fields defined by $h'_{\mu\nu}$ in Equation (81). This is possible again by using analogous results as those reported in Equations (84) and (85). The explicit result is

$$\frac{\partial}{\partial x^\nu} h_{00vac} = 4A(\alpha)T_0'(r,t)T_0''\delta^\nu{}_r, \qquad\qquad \frac{\partial^2}{\partial x^\kappa \partial x^\mu} h_{rrvac} = 4B(\alpha)T_0'(r,t)T_0''\delta^\nu{}_r,$$

$$\frac{\partial^2}{\partial x^\kappa \partial x^\mu} h_{0rvac} = \frac{T_0''(r,t)[-2(1+\alpha)^7 + \alpha[1+3\alpha(1+\alpha)][2+\alpha(6+\alpha(6+\alpha))]]}{2(1+\alpha)^4(1+3\alpha(1+\alpha))}\delta^\nu{}_r. \tag{87}$$

Note that by considering $T_0'(r,t)$ infinitesimally close to zero, it is possible to ignore some of the previous contributions. What is important to notice at this level is that the terms $\Gamma\Gamma$ for the second part of the Riemann tensor definition in Equation (82) can be considered to vanish for a free-falling frame in GR. However, in massive gravity, at the perturbative level, we have to retain the contributions coming from the Stückelberg functions when we consider the case $\beta = \alpha^2$. In this case, it is evident that the term $\Gamma\Gamma$ will be equivalent to the gauge-field which provides the graviton mass dynamically. It is analogous to the photon field term $(A_\mu)^2$ in Equation (48). The connection $\Gamma$ is an explicit function of the derivatives of the Stückelberg function. The contractions in order to derive the Ricci scalar, for the kinetic term given in Equation (83), can be done with the conformal Minkowski metric $ds^2 = S_0^2 ds_M^2$ given by Equation (58) without considering the Stückelberg function. This is correct since here we are assuming $T_0'(r,t)$ to be infinitesimally close to zero. For the case of the connection terms, if we make the relevant contractions and after expanding for small values of $T_0'(r,t)$, the final form of the Lagrangian is

$$\mathcal{L} = \mathcal{L}_{EH} + F(v,\alpha)(\Gamma\Gamma)_{gauge} + V(g,\phi), \tag{88}$$

where $\mathcal{L}_{EH}$ is the Einstein-Hilbert Lagrangian, $F(v,\alpha)$ is a function of the vacuum parameter defined in Equation (78). This term represents the expansion around the false vacuum and it is responsible for the appearance of the massive term. The $\Gamma$ matrices correspond to the portion of the Christoffel connection which depends explicitly on the Stückelberg functions at the perturbative level. Here we can see that the graviton mass comes dynamically from the gauge fields $\Gamma$. For a vanishing spatial dependence of $T_0(r,t)$, this term vanishes in a free-falling frame. This term will only appear for observers defining

the time arbitrarily with respect to $T_0(r,t)$ (connected to the case $\beta = \alpha^2$). The observers defining frame of references in agreement with $T_0(r,t)$, will not be able to detect the dynamical massive term neither. This situation is analogous to a spontaneous magnetization below certain temperature in the ferromagnetism phenomena. Note that in addition, the spatial variation of $T_0(r,t)$ is the responsible for the appearance of the function depending on $v$. $V(g,\phi) = \sqrt{-g}U(g,\phi)$ is the potential expanded up to second order in perturbations and it might contain some mix terms that can be absorbed after some trivial transformations.

*Generic Character of the Dynamical Origin of the Graviton Mass*

Although all the derivations in this paper were done inside the scenario of the dRGT theory of massive gravity, it can be demonstrated that the result obtained in (88) is a generic one. We can explore, for example, the model proposed by Chamseddine and Mukhanov and its extensions formulated by Oda in [38,39]. The extensions in [39] model suggest an action of the form

$$S = \frac{1}{16\pi G} \int d^D x \sqrt{-g} \left( R - V(H^{AB}) \right), \tag{89}$$

which is analogous to the Equation (1) if we identify the potential term $U(g,\phi)$ with $V(H^{AB})$ in the previous equation. $V(H^{AB})$ is a non-derivative interaction term in the action. Here $H^{AB}$ is defined as

$$H^{AB} = \eta^{AB} - h^{AB} + \partial^A \varphi^B - \partial^B \varphi^A. \tag{90}$$

Here $\eta^{AB}$ is a Minkowski metric with the internal index notation. Furthermore, $\varphi^A$ are the fluctuations of the Stückelberg fields $\phi^A = x^\mu \delta^A_\mu + \varphi^A$. This notation is standard in any theory of massive gravity. If we translate the Stückelberg fields inside the dynamical metric, then we would obtain definitions analogous to those found in Equation (15). The vacuum condition for the previous potential term, suggests that [39]

$$\frac{\partial V(H^{AB})}{\partial H^{AB}} = \frac{1}{2}\eta_{AB}V(H_*) = 0. \tag{91}$$

By applying the Euler-Lagrange equations, it is not difficult to find the equations of motion

$$\nabla^\mu \left( \frac{\partial V}{\partial H^{AB}} \nabla_\mu \phi^B \right) = 0,$$

$$R_{\mu\nu} - \frac{1}{2}g_{\mu\nu}R = \frac{1}{2}V(H_*) \left( \frac{1}{2}\eta_{\mu\nu} - h_{\mu\nu} \right) - \frac{\partial^2 V(H_*)}{\partial H^{\mu\nu}\partial H^{\rho\sigma}}. \tag{92}$$

Here the first equation represents the energy-momentum conservation in analogy with the work developed in [9]. The second equation represents an extension of Einstein's equations. We can perceive from this second equation that the left-hand side is the result of the variation of the standard Einstein-Hilbert action. This result would correspond to the variation of the term $\pounds_{EH}$ in Equation (88). The first term on the right-hand side in Equation (92) would naturally vanish in a well-defined vacuum and it would correspond to the term $V(g,\phi)$ in Equation (88). Finally, the second term on the right-hand side of Equation (92) corresponds to the term $F(v,\alpha)(\Gamma\Gamma)_{gauge}$ in Equation (88). This demonstrates the generic character of the formulation developed in this paper. In any formulation of massive gravity, we can repeat the same steps and find similar conclusions. This is the case because the massive terms in general contain a potential term containing non-derivative-terms interactions in the action and normally, these additional terms enter inside the action as a pair (of terms) introducing two free-parameters in the theory. Interestingly, whether the vacuum is degenerate or not will depend on the relation between the two free-parameters introduced in this way [6].

## 9. Conclusions

In this paper we have explained the dynamical origin of the graviton mass. We have explained the analogy between the massive gravity case with the standard formulation of the Higgs mechanism in particle physics. Our formulation suggests that for some combination of the free-parameters of the theory, the vacuum has a preferred time direction and then it becomes degenerate. Then the symmetry is spontaneously broken and any generator related to the time-direction is broken. In this paper we have worked under the stationary condition assumption but an extension to more general situations is trivial. For purposes of analysis we have also assumed that the deviations of the Stückelberg function with respect to the ordinary time-coordinate are small. We have worked on a free-falling frame of reference such that the effects of the gravitational sources can be ignored locally. We have demonstrated that the symmetry is spontaneously broken for the combination of parameters satisfying the condition $\beta = \alpha^2$. The order parameter corresponds in general to the total derivative $dA(r, t)$, which under the stationary condition assumption, becomes equivalent to $T_0'(r, t)$. In massive gravity we have three Nambu-Goldstone bosons, which are eaten-up by the dynamical metric, becoming then massive. When we analyze this scenario by expanding the action up to second order, we find that a fraction of the gravitational connections correspond to the gauge-fields of the theory absorbing the extra-degrees of freedom. It is not strange to see the coincidence between a mechanism generating the mass dynamically in massive gravity and the standard Higgs mechanism of particle physics since it has been demonstrated that massive gravity behaves as a gravitational $\sigma$-model [5,7,8]. Alternative formulations for the dynamical origin of the graviton mass has been done in [40]. In [38,39], the graviton Higgs mechanism was analyzed by using a theory different from the dRGT massive gravity. Here we have demonstrated that our formulation is generic and it can be applied to any formulation of massive gravity, including the formulations done in [38,39].

**Funding:** This research was funded by the JSPS fellow for oversea researchers, grant Number P15322 and by the CAS PIFI program.

**Acknowledgments:** The author would like to thank Petr Horava and John Ellis for useful discussions.

**Conflicts of Interest:** The author declare no conflict of interest.

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
