# Peer review of "The Dynamical Origin of the Graviton Mass in the Non-Linear Theory of Massive Gravity"

_universe, doi:10.3390/universe5070166_

Round 1

Reviewer 1 Report

The present manuscript investigates the dynamical origin of the graviton mass in the context of dRGT theory of massive gravity.
After an introductory review on the the dRGT formulation of massive gravity and the standard Higgs mechanism for scalar fields, the author analyzes the massive action at the background and perturbation levels. In doing so, he studies the mechanism generating the graviton mass through the Nambu-Goldstone theorem.

The subject matter is interesting and paper is well written. However, the following issues should be addressed before I can recommend the paper for publication:

1) Foremost, all the figures are missing. I recommend the author to fix this unfortunate problem in order to make the discussion of the results clear for the reader.

2) The whole analysis is carried out in the dRGT framework. The author should explain why his results do not depend on the particular theory of massive gravity used.

3) The author should discuss his results in view of alternative formulations for the origin of the graviton mass developed so far in the literature. In particular, I believe that a brief comparison among the outcomes of the various theories is necessary.

After these changes, I would like to see the manuscript again to take my final decision.

Author Response

I thank the referee for his positive comments and suggestions. They have helped me to improve the content of the paper. My responses to the referee comments are as follows:

Referee #1 comment:

The present manuscript investigates the dynamical origin of the graviton mass in the context of dRGT theory of massive gravity. After an introductory review on the the dRGT formulation of massive gravity and the standard Higgs mechanism for scalar fields, the author analyzes the massive action at the background and perturbation levels. In doing so, he studies the mechanism generating the graviton mass through the Nambu-Goldstone theorem. 

The subject matter is interesting and paper is well written. However, the following issues should be addressed before I can recommend the paper for publication:

1)     Foremost, all the figures are missing. I recommend the author to fix this unfortunate problem in order to make the discussion of the results clear for the reader.

My response:

Thanks to the referee for this advice. Following the referee suggestions, I have verified that the figures appear in the pdf version.  

Referee #1 comment:

2) The whole analysis is carried out in the dRGT framework. The author should explain why his results do not depend on the particular theory of massive gravity used.

My response:

I thank the referee for this important comment. Following the referee suggestions, I have extended the paper explaining why the obtained results are generic and not only related to the dRGT massive gravity theory. For that purpose, I have included a new subsection (please read it at the end of section (8)). In this new sub-section, I explain how the results of this paper can be extended to the analysis of other massive gravity theories and I explain how the compact result defined eq. (88) can be compared term by term with the results obtained in the new added references [21] and [22], which consider a theory different to the dRGT one.    

Referee #1 comment:  

3) The author should discuss his results in view of alternative formulations for the origin of the graviton mass developed so far in the literature. In particular, I believe that a brief comparison among the outcomes of the various theories is necessary.

After these changes, I would like to see the manuscript again to take my final decision.

My response:

I thank the referee for these comments. Following this suggestion, since this comment is an extension of the previous one, I want to remark here that in this new version of the paper, the referee will find in the new added subsection (at the end of section (8)), a comparison with other models. Since all the theories of massive gravity introduce non-derivative terms interactions in the action in similar fashion, the added comparison, illustrates the generic character of the analysis contained in this paper. I want to remark that I have added two new references, relevant for the mentioned comparison and I have added some text at the end of the conclusions.

Once again I thank the referee for the positive comments and for the suggestions which helped to improve the content of the paper.  

Reviewer 2 Report

The authors have studied the dynamical origin of the graviton mass in dRGT massive gravity theory. A comparison is done with the Higgs mechanism with a scalar field as to the generation of graviton mass. The paper is well written and I recommend the paper for publication.

Author Response

Referee #2 comment:  

The authors have studied the dynamical origin of the graviton mass in dRGT massive gravity theory. A comparison is done with the Higgs mechanism with a scalar field as to the generation of graviton mass. The paper is well written and I recommend the paper for publication.

My response:

I thank the referee for this positive comments and for the acceptance of the paper.

Round 2

Reviewer 1 Report

The author addressed all the points of my previous report and improved the quality of the paper.

Therefore, I recommend the manuscript for publication.